# Differential Expression of Proteins in an Atypical Presentation of Autoimmune Lymphoproliferative Syndrome

**DOI:** 10.3390/ijms23105366

**Published:** 2022-05-11

**Authors:** Dulce María Delgadillo, Adriana Ivonne Céspedes-Cruz, Emmanuel Ríos-Castro, María Guadalupe Rodríguez Maldonado, Mariel López-Nogueda, Miguel Márquez-Gutiérrez, Rocío Villalobos-Manzo, Lorena Ramírez-Reyes, Misael Domínguez-Fuentes, José Tapia-Ramírez

**Affiliations:** 1Unidad de Genómica, Proteómica y Metabolómica, Laboratorio Nacional de Servicios Experimentales (LaNSE), Centro de Investigación y de Estudios Avanzados, Mexico City 07360, CP, Mexico; cdelgadillo@cinvestav.mx (D.M.D.); eriosc@cinvestav.mx (E.R.-C.); lramirez@cinvestav.mx (L.R.-R.); jom_sayap@hotmail.com (M.D.-F.); 2Unidad Médica de Alta Especialidad (UMAE), Centro Médico Nacional La Raza Hospital General, Mexico City 02990, CP, Mexico; adriana.cespedes@imss.gob.mx (A.I.C.-C.); reumapedia@gmail.com (M.G.R.M.); reuma.pediatrica@gmail.com (M.L.-N.); miguel.marquez@imss.gob.mx (M.M.-G.); 3Departamento de Genética y Biología Molecular, Centro de Investigación y de Estudios Avanzados, Mexico City 07360, CP, Mexico; rvillalobos@cinvestav.mx

**Keywords:** apoptosis, autoimmune lymphoproliferative syndrome, proteomics

## Abstract

Autoimmune lymphoproliferative syndrome (ALPS) is a rare disease defined as a defect in the lymphocyte apoptotic pathway. Currently, the diagnosis of ALPS is based on clinical aspects, defective lymphocyte apoptosis and mutations in *Fas*, *FasL* and *Casp 10* genes. Despite this, ALPS has been misdiagnosed. The aim of this work was to go one step further in the knowledge of the disease, through a molecular and proteomic analysis of peripheral blood mononuclear cells (PBMCs) from two children, a 13-year-old girl and a 6-year-old boy, called patient 1 and patient 2, respectively, with clinical data supporting the diagnosis of ALPS. *Fas*, *FasL* and *Casp10* genes from both patients were sequenced, and a sample of the total proteins from patient 1 was analyzed by label-free proteomics. Pathway analysis of deregulated proteins from PBMCs was performed on the STRING and PANTHER bioinformatics databases. A mutation resulting in an in-frame premature stop codon and protein truncation was detected in the *Fas* gene from patient 2. From patient 1, the proteomic analysis showed differences in the level of expression of proteins involved in, among other processes, cell cycle, regulation of cell cycle arrest and immune response. Noticeably, the most down-regulated protein is an important regulator of the cell cycle process. This could be an explanation of the disease in patient 1.

## 1. Introduction

Programmed cell death, or apoptosis, plays a critical role in regulating lymphocyte development and homeostasis in the immune system. Apoptosis defects may contribute to abnormal lymphocyte accumulation, autoimmunity, and lymphoid malignancy. B- and T-lymphocyte apoptosis are initiated by binding of the Fas ligand to Fas, a transmembrane molecule belonging to the tumor necrosis factor receptor I (TNFR I) gene family. Fas transduces the death signal through its highly conserved cytoplasmic “death domain”, which represents the binding site for proteins that activate cysteine proteases called caspases. These caspases undergo proteolytic autoprocessing and cleave, in a signaling cascade, downstream effector caspases and other targets leading to apoptosis [1,2].

Autoimmune lymphoproliferative syndrome (ALPS) is a rare disease defined as a defect in the lymphocyte apoptotic pathway and characterized by chronic massive, nonmalignant lymphadenopathy, splenomegaly and/or hepatomegaly; expansion of a normally rare population of T cells bearing ab-antigen receptors but lacking both CD4 and CD8 coreceptors (α/β double-negative T cells, α/βDNTs) and defective in vitro apoptosis of mature lymphocytes [3,4]. Patients with ALPS symptoms have been genetically characterized, and most (60–70%) harboring mutations on the *Fas* gen are classified as ALPS-FAS. With less frequency (<1%), mutations have been found on the *Fas* ligand gene (ALPS-FASL). Patients (2–3%) harboring mutations in the *caspase 10* gene (*Casp10*) are classified as ALPS-CASP10. However, not all individuals carrying mutations on those genes show overt disease or atypical phenotype; even more, 20–30% of the patients clinically diagnosed do not present any defined mutations and are classified as undetermined (ALPS-U) [5,6,7,8]. Elevated biomarkers such as plasma soluble Fas ligand (sFASL), IL-10, IL-18 and plasma or serum vitamin B-12 have been used to presumptively diagnose ALPS [9]; however, these biomarkers may also be elevated in other conditions, including common variable immunodeficiency [1]. In spite of the mutations documented and even the clinical aspects reported by patients, ALPS has been misdiagnosed [5,10].

In Mexico, there is no record of how prevalent ALPS is, and, until 2017, the General Health Council has updated a list of 14 rare diseases that are prevalent in the country; however, ALPS is not included (http://www.csg.gob.mx/; accessed on 27 April 2018). To go one step further in the knowledge of the illness, we performed a molecular and proteomic analysis of two patients who fill the clinical diagnosis criteria of ALPS. For a molecular diagnostic, in both patients we search for mutations in the genes involved in the disease. As a result of this diagnosis, one patient was defined as ALPS-U, and a total protein sample from her was used for the proteomic study. Proteomics, including the identification and comparison of a repertoire of proteins in a specific cellular state, should help discover molecules that play critical roles in regulating T-cell proliferation and provide significant information for biological problems such as those derived from this or any other pathological process [11]. The information achieved will provide us with valuable elements to make a more integral diagnosis of ALPS. In addition, this is the first proteomic analysis of human T lymphocytes derived from a Mexican case with a clinical diagnosis of ALPS.

## 2. Results

### 2.1. Clinical Features of the Patients

ALPS is a rare disease defined as a defect in the lymphocyte apoptotic pathway, and in Mexico, it is not considered in the list of 14 rare diseases prevalent in the country. In fact, no record exists of its incidence. We performed this study with two kids, a 13-year-old girl and a 6-year-old boy, called patient 1 and patient 2, respectively; both patients showed data to support a diagnosis of ALPS because of a recent history characterized by recurrent abdominal pain, asthenia, adynamia, jaundice and hepatosplenomegaly, in addition to leucopenia, thrombocytopenia, hemolytic anemia, negative ANA and anti-DNA antibodies, facial erythema, epistaxis and an increase in the number of double negative CD4/CD8 cells (Table 1). A liver biopsy from patient 1 showed damage compatible with autoimmune hepatitis besides the presence of anti-smooth muscle antibodies (1:40). Due to all these data, the patients were medicated with immunosuppressive agents during years of clinical monitoring. Although the levels of lymphocytes diminished, keeping values in a normal range, both kids presented high levels of immunoglobulins. As it is known, the activation of B-cell lymphocytes is, in turn, responsible for the production of immune complexes and autoantibodies [12] and can conduce to the development of an autoimmune disease.

### 2.2. Molecular Analysis of Fas, FasL and Casp10 Genes

Due to the fact that mutations in *Fas*, *FasL* and *Casp10* genes have been reported to be associated with ALPS [1,13,14], we obtained the coding sequences of these genes through the isolation of RNA from PBMCs of the patients. The isolated RNA was used as a template in RT-PCR assays. Sequences of sense and antisense cDNA chains were analyzed and translated using the ExPASy Translate tool (http://expasy.org/translate/; accessed on 24 April 2019). In the nucleotide sequence for the Fas gene from patient 2, we found a C to T transition in an allele (Figure 1). This change has been reported [15], resulting in an in-frame premature stop codon and protein truncation at residue 234 (R234•) in a highly conserved region of the Fas death domain. No mutations were found in the *Fas* gene in patient 1 (Table 2).

While seven Caspase 10 variants have been described as showing a high heterogeneity in the COOH terminus [16], for that reason, we performed the alignment of the sequences from the patients with the coding sequence of the Caspase 10 variant A, considered as the canonical (Table 3). Finally, the sequence of FasL from the patients was also aligned with the consensus (Table 4). No mutations were found in *Casp10* and *FasL* genes from the patients.

### 2.3. Proteomic Analysis of PBMCs

Since a genetic cause of ALPS was discarded in patient 1, she was called ALPS-U, and we focused our work on the knowledge of the proteins present in her PMBCs. As it is known, the protein expression and post-translational modifications profiled by proteomics can explore functional interactions between proteins and develop disease-specific biomarkers by integrating with clinical information [22]. Thus, a quantitative proteomic analysis was performed. As a control, PBMCs proteins from a 13-year-old healthy girl were used.

Firstly, our analysis of reliability and confidence at peptide/protein level based on Figures of Merit (FOMs) for label-free proteomic studies by Souza et al. [23] showed a proper mass spectrometer calibration, efficient enzymatic digestion, adequate sensibility and correct normalization of injection demonstrating that both samples were comparable to each other (Appendix A). Consequently, we were capable of detecting in UDMS^E^ mode a total of 116081 peptides summarized in all injections (Appendix A), with a dynamic range ~6 orders of magnitude, expressed as log_10_ (Appendix A).

In this study, 1521 proteins were detected. Of those, 1399 were quantified, and 122 were present in either the control or ALPS condition. We applied to all proteins a filter that consisted of a coefficient of variation (CV) ≤ 0.15; at least two peptides per protein; at least one unique peptide and ANOVA (*p*) ≤ 0.05. Total false positives (reversed proteins) were eliminated. Finally, 266 highly reliable quantified proteins were scattered in a volcano plot (Figure 2), where 77 and 34 were detected as up- and down-regulated, respectively (Appendix A). Besides this, we detected proteins that were exclusively expressed in the PBMCs either from the control and from the patient (35 and 16 proteins, respectively) (Appendix A). All results are summarized in Appendix A.

### 2.4. Identification of Biological Pathways of Detected Proteins

The distribution pattern of proteome-wide changes in PBMCs from the analyzed patient (ALPS-U) is shown in Figure 3. The functional characterization of identified proteins was based on Gene Ontology (GO) using Panther Classification System bioinformatics software to obtain information about their molecular function. The classification revealed that most proteins possessed the ability to bind (49.2%) and were involved in catalytic activities (29.8%) (Figure 3a).

For their participation in binding processes, most of the proteins were grouped in protein binding (63.9%) and heterocyclic compound binding and organic cyclic compound binding (45.9% each) (Figure 3b). HSP90AA1, HSP90AB1, HBB, HBG2, HBA2, HBA1, ARHGDIB, RPS27A and E2F7 were present in the three mentioned groups and, interestingly, except E2F7, which is down-regulated, the rest are up-regulated. Moreover, 21 proteins form the group involved in heterocyclic compound binding (Figure 3c), and, from these, 19 are up-regulated, and only 2 are down-regulated. Noticeably, these 2 down-regulated proteins are implicated in the nucleic acid binding as well as 8 of the 19 up-regulated proteins. This group of 21 proteins covers molecular activities from mRNA metabolism as SYNCRIPT; oxygen transport as HBD, HBB, HBA1 and HBA2 through the tetrapyrrole binding; as components of the nucleosomes such as HIS1H2BD and HIST1H2AH, that, like proteins such as ORC2, play a central role in transcription regulation, DNA repair, DNA replication and chromosome stability. Additionally, also as part of this group, CRLF3 and E2F7 are involved in cell cycle control as well as HSP90AA1 and HSP90AB1 through nucleic acid and nucleoside phosphate binding, respectively (Appendix A) (https://www.genecards.org/; accessed on 12 July 2019).

On the other hand, a group of 20 proteins was classified as involved in catalytic activities; from these, 15 are up-regulated, and 5 are down-regulated. Three of the five down-regulated, the kinesins-like proteins KIF18A and KIF22 as well as the D-aminoacyl-tRNA deacylase 1, DTD1, are implicated in the cell cycle. These kinesins act on the formation and movements of the chromosomes, and DTD1 binds the DNA unwinding element and plays a role in the initiation of DNA replication. In the same sense, 6 of the 15 up-regulated proteins from this group participate in different points of the cell cycle as RPA1, which stabilizes ssDNA intermediates during DNA replication or upon DNA stress. This prevents their reannealing and recruits and activates proteins and complexes involved in DNA metabolism; MHL3 is implicated in maintaining genomic integrity during DNA replication and after meiotic recombination; the tyrosine-protein kinase ITK, involved in T-cell proliferation and differentiation, and TLK1, which participates in the regulation of chromatin assembly (Appendix A) (https://www.genecards.org/; accessed on 12 July 2019).

We also performed an in-silico analysis using Panther software to compare the participation of differentially regulated proteins in biological pathways. Interestingly, 12 of 14 biological processes gathered most of the up-regulated proteins (Figure 4).

### 2.5. Protein-Protein Interactions

STRING network analysis was used to elucidate interactions among all the proteins identified in the proteomic study; this is differentially regulated as well as those expressed only in ALPS-U or in the control samples. According to the STRING algorithm, in our assay, the proteins are, as a group, at least partially biologically connected. As previously mentioned, ALPS is characterized by a defect in the lymphocyte apoptotic pathway, so we looked for eight biological processes related to the disease as regulation of the immune system process, immune response, cell cycle and cell cycle process, regulation of cell cycle arrest as well as regulation of cell death, regulation of apoptotic process and positive regulation of cell death. In this way, we found 62 proteins involved in the mentioned processes (Appendix A). From them, 40 and 10 were detected as up- and down-regulated, respectively (Appendix A). At the same time, the rest of the 12 proteins were present exclusively, either in the control or ALPS-U condition (8 and 4, respectively) (Appendix A).

In the group of up-regulated proteins, RHOA and THBS1 participate in seven of the eight biological processes, only excluding the regulation of cell cycle arrest (Appendix A). This is relevant because, according to the Gene Cards Human Gene Database, overexpression of RHOA is associated with tumor cell proliferation and metastasis, while THBS1 plays a role in platelet aggregation, angiogenesis, and tumorigenesis. On the other hand, CFL1 and LMNA share the same four molecular processes, whereas NCK2 and HSP90AA1 are also involved in four activities, although they share only one: regulation of the immune system process. However, it is very well known that the chaperone HSP90AA1 is also involved in a plethora of biological processes due to its aid in the proper folding of specific target proteins (https://www.genecards.org/; accessed on 30 July 2019). Several hundred proteins are substrates or clients of HSP90AA1, including diverse key signaling pathway proteins, transcription factors and cell cycle proteins. This makes HSP90AA1 essential in eukaryotes and a central modulator of important processes that range from stress regulation and protein folding to DNA repair, development, the immune response, neuronal signaling, signal transduction and, interestingly, cell cycle control, cell proliferation and differentiation, among many others processes [24]. In our analysis, we found a direct interaction of HSP90AA1 with 21 proteins; from those, only 6 are not involved in the processes of our interest, while 13 were detected as up-regulated, 4 down-regulated, 1 expressed only in the control and 3 exclusively in the ALPS-U condition. In the same group of up-regulated proteins and regarding immune response and regulation of immune system process, we identified 31 proteins, whereas, in processes such as regulation of cell death, regulation of the apoptotic process and positive regulation of cell death resulted in 18, 16 and 12 proteins, respectively (Appendix A).

Another interesting protein was LGALS1, a galectin that plays a role in regulating apoptosis, cell proliferation and cell differentiation. LGALS1 interacts with TF, which participates in the transport of iron from sites of absorption and heme degradation to storage and utilization sites and may also have a further role in stimulating cell proliferation (Appendix A) (https://www.uniprot.org/; accessed on 12 December 2021). It is interesting that, in the disease condition, TF is reduced 2.59 times (Log_2_ Fold change −1.37) while LGALS1 is exclusively present in the control (Appendix A). This could explain the anemia of the patient because of the excess lymphocytes and the decrease in the transport of iron.

Given the high number of proteins detected in our assays and, to clarify the interaction between the proteins involved in the cell cycle, regulation of cell cycle and cell cycle process, we performed another STRING network analysis. In this case, only proteins that participated in these mentioned processes were included (Figure 5). In addition to the previously mentioned, seven other up-regulated proteins are involved in cell cycle and cell cycle process, MHL3, PCNT, CRLF3, CECR2, ORC2, RPA1 and TLK1, and, from those, CRLF3 also participates in the regulation of cell cycle arrest. Although it did not interact with any of the selected proteins used to perform the network of Figure 5, it was detected as up-regulated. In fact, MLH3, PCNT and CRLF3 present a fold change of 42.44138, 26.88151 and 14.13389, respectively (Log_2_ Fold change 5.4074, 4.748542 and 3.821086, respectively) being the most up-regulated proteins in our assay with participation in the indicated processes. As was indicated, CRLF3 is a DNA binding protein nucleotide phosphatase that may negatively regulate cell cycle progression at the G0/G1 phase. MLH3 belongs to a family of DNA mismatch repair (MMR) genes implicated in maintaining genomic integrity during DNA replication and after meiotic recombination. While PCNT is involved in mitotic spindle organization, DNA damage checkpoint and primary cilia formation, it is crucial for mitotic spindle pole formation and disintegration (https://www.genecards.org/; accessed on 12 December 2021) (Figure 5).

Furthermore, in the group of down-regulated proteins, as mentioned, only 10 proteins are involved in the processes studied. E2F7, the most down-regulated protein detected in the assay (Log_2_ Fold change −4.65302, approximately 50 times less when was expressed as base 10 logarithm compared to the control), belongs to this group. In addition, E2F7 interacts with two more down-regulated proteins, KIF18A and KIF22, members of the kinesin family responsible for the movement along microtubules and, therefore, with an essential role in metaphase chromosome alignment and maintenance (Appendix A, Figure 5) (https://www.genecards.org/; accessed on 12 December 2021). More stirring is that E2F7 is a member of the E2F family of transcription factors whose function is to activate and/or repress the transcription of many essential genes involved in cell proliferation, apoptosis and differentiation [25,26]. This becomes fascinating because E2F7 directly represses the expression of specific G1/S genes involved in DNA replication, metabolism and repair; thus, E2F7 controls S-phase progression by repressing E2F target genes [27]. Moreover, in a detailed analysis and according to the web-based atlas at www.targetenereg.org, (accessed on 2 September 2019) [28], our results show four proteins involved in the G1/S phase, two of these are up-regulated (HSP90AA1 and ORC2), and two are down-regulated (ANK3 and E2F7) (https://www.genecards.org/; accessed on 2 September 2019). HSP90AA1 and E2F7, aside from HIST1H2AH, which is essential in nucleosome formation, are regulated by members of the E2F family, as well as the down-regulated KIF18A, although this participates in the G2/M phase. In our network, E2F7 and the kinesins KIF18A and KIF22 are joined to the rest of the proteins through CORO1A, a protein involved in cell cycle progression, signal transduction, apoptosis, and gene regulation, detected as up-regulated (Log_2_ Fold change 1.755302).

### 2.6. E2F7 Study

Finally, we sought to independently validate the expression of genes indicated by label-free mass spectrometry. The transcription factor E2F7, which was identified by three peptides (Figure 6), showed a severe decrease in its expression. The dynamic expression range of the quantified protein showed a difference of approximately 50 times less in the sample of patient 1 (ALPS-U) compared to the control when it was expressed as base 10 logarithm (Log_2_ Fold change −4.65302) (Figure 7a). On the other hand, CRLF3, a DNA binding protein nucleotide phosphatase that may negatively regulate cell cycle progression at the G0/G1 phase, was detected as the fifth most up-regulated protein (Log_2_ Fold change 5.4074). The expression of E2F7 and CRLF3 was examined by RT-PCR (Figure 7b). No differences were found at the transcription level for both genes. Validation of E2F7 and CRLF3 at the protein level was performed by Western blot (Figure 8). It is known that changes in protein levels are not accompanied by changes in corresponding mRNAs and that a single genetic perturbation leads to progressive widespread changes in several molecular layers [29]. In fact, gene expression is one of the most fundamental processes in biology, and it involves four fundamental cellular processes, transcription, mRNA degradation, translation, and protein degradation. In addition, around 40% of the variability in protein levels in mouse fibroblasts has been explained by the mRNA levels [30]. In our assay, we also included a protein sample from the boy patient who presented the genetic alteration in the *Fas* gene; we could verify, in both patients, the decrease in the expression of E2F7 and the increase of CRLF3 (Figure 8), which suggested that the difference in the levels of these proteins may be a consequence of the deregulation from the disease itself, regardless of its origin.

## 3. Discussion

T-lymphocyte activation and proliferation take place in a highly coordinated program to ensure the acquisition of optimal effector functions and cell survival. Owing to the central roles of T cells in many autoimmune and non-autoimmune disorders, a great amount of work has been done on identifying molecules that manipulate T-cell proliferation and function [11,31].

Rare diseases are difficult to study due to the lack of agreement in their definition and the fact that they do not have anything in common apart from their rarity; this affects patient resources, diagnosis, and treatment approaches, as well as research on them and potential therapies. In addition, even taking into account the knowledge of a genetic cause in some rare diseases, their diagnosis is often delayed or wrong, and optimal clinical management is seldom achieved [32]. Technologies such as proteomics have emerged, providing powerful tools to scan the realm of expressed proteins and investigate perturbing pathways causing diseases as well as the identification of biomarkers that can be used for diagnosis and prognosis or in monitoring the efficacy of pharmacological treatments [33]. Although so far descriptive, some proteomic studies have been reported as an alternative to approach rare diseases. Sjögren syndrome, for example, has been studied by combining two-dimensional electrophoresis and matrix-assisted laser desorption/ionization (MALDI-TOF/TOF) mass spectrometry (MS) comparing salivary samples from a patient before, during and after pharmacological therapy. Results from this study suggest a correspondence between the patient’s clinical improvement and the changes in their proteomic salivary profile [34]. In 2017, Aqrawi et al. [35] proposed liquid chromatography-mass spectrometry (LC-MS) as a potential non-invasive diagnostic tool to detect biomarkers on fluids from patients affected by this syndrome. These authors detected up-regulated proteins involved in innate immunity, cell signaling, wound repair, adipocyte differentiation and B-cell survival. On the other hand, in a heterogeneous spectrum of rare human diseases characterized by alterations in the LMNA gene encoding the nuclear envelope proteins lamins A/C and collectively denominated laminopathies, MALDI-TOF/MS and Western blot were used to suggest that LMNA defects affect primarily the expression of proteins involved in cytoskeleton organization, energy metabolism and oxidative stress response [36].

ALPS is a rare disease resulting from the non-functional or dysfunctional apoptosis of lymphocytes. It has been proposed that a presumptive diagnosis of ALPS can be made by detecting high levels of αβDNTs cells and a definitive diagnosis by identifying mutations in the relevant apoptosis genes (*Fas*, *FasL* or *Casp10*). Detection of elevated levels of serum vitamin B-12, sFASL, IL-10 and IL-18 also have been proposed as part of the diagnostic criteria. Nevertheless, due to an extended gamut of symptoms, the disease has been misdiagnosed [6,37]. In 1998, Infante et al. reported the study of a family of a kindred containing 11 individuals of four generations carrying a mutation in the intracellular death domain of the *Fas* gene, resulting in failure to bind to FAS-associated death domain protein, an important mediator of the apoptosis signal [38]. Monitored for as long as 25 years, this family allowed longitudinal and cross-sectional observations of the clinical spectrum of ALPS. In that way, a conclusion of the report is the variability in clinical presentation among individuals documented to carry the same unique *Fas* mutation. This is because three persons of the studied family bear *Fas* mutation and do not present symptoms of ALPS, while other family members display a wide range of clinical and laboratory abnormalities [38].

Besides the difficulty of making a correct diagnosis of the disease, in Mexico, there are no studies about the incidence of ALPS.

Here, we report the cases of two kids, a girl and a boy, with clinical and immunological criteria for a probable diagnosis of ALPS [9]. The DNA sequences of the genes involved in the disease (*Fas*, *FasL* and *Casp10*) were reviewed thoroughly in both patients. A transversion of C to T in an allele of the *Fas* gene from the boy patient was detected, resulting in an in-frame premature stop codon. This change located in a highly conserved region of the Fas death domain has been reported to be associated with ALPS [15]. Fas transduces the death signal through its highly conserved cytoplasmic death domain, thereby initiating programmed cell death. Fas is expressed on thymocytes and activated T and B cells and is thought to be primarily responsible for the apoptosis of antigen-primed, activated lymphocytes, so a defective function could cause an accumulation of lymphocytes, including potentially autoreactive cells [15].

On the other hand, no changes were detected in the nucleotide sequence of any of the genes involved in the disease in the sample of the girl patient. Intending to go further in the knowledge of the disorder in this person, we decided to perform a proteomic analysis. For that, in the present study, we have used a label-free MS approach to reveal qualitative and quantitative proteomic differences between PBMCs from the ALPS-U patient and a healthy girl of the same age. Even though most of the proteome is shared between the two girls, we observed clear differences in protein abundance and a significant differential expression that might have functional effects on the physiology of the patient affected with this rare disease. The total number of detected and quantified proteins were analyzed in silico by their molecular function. Results show that almost 50% participate in processes of binding to heterocyclic and organic cyclic compounds. We focused on these two processes due to both cover important biological ways for the cell cycle as nucleic acid binding and nucleoside binding and for the cell metabolism through the tetrapyrrole binding.

Considering deregulation in the cell proliferation process as a typical feature of the disease, we detected at least 20 deregulated proteins involved in the cell cycle, cell cycle process and regulation of cell cycle arrest. Among these proteins, MLH3 and PCNT showed the highest values of expression (42 and 26-fold, respectively) followed by CRLF3, CECR2, ORC2, RPA1 and LMNA (14 to 4.96-fold), and in less dimension HSP90AA1, RHOA, CFL1, TLK1 and THBS1 (2.97 to 2.48-fold). It is interesting that four of these proteins (LMNA, PCNT, TLK1 and CRLF3) participate in the G2/M phase of the cell cycle. Laminin and Pericentrin are integral components of the filamentous matrix involved directly in nuclear and cytoplasm stability, along with other proteins also up-regulated (such as Vim, for example, which functions as an organizer of proteins). As such, it is important to mention that the serine/threonine kinase 1 (TLK1) participates in the regulation of chromatin assembly and helps to maintain its structure. Together, these proteins participate in preparing the cell for division. Additionally, the Cytokine Receptor-Like Factor 3 (CRLF3) may negatively regulate cell cycle progression at the G0/G1 phase (https://www.genecards.org; accessed on 28 March 2022).

This certainly makes huge deregulation evident in the mentioned processes. In addition to this, we must point out the detection of Caspase 3. A key protein involved in the process of apoptosis was found to be 2.23483-fold less expressed in ALPS, but due to the strict parameters used, it did not pass our filter since it presented a coefficient of variation above 40% (Appendix A). In the process of patient care, a splenectomy was performed, and as a result, it was not possible to obtain a sample that would allow us to continue with the reliable follow-up of the expression of the proteins involved. The activity of Caspase 3, even in small amounts, could be induced by the drugs used in the treatment of the patient, specifically by the action of mycophenolate, for which effects on lymphocytes have been proved [39]. However, it looks not enough to maintain the balance between cellular proliferation and apoptosis.

Interestingly, E2F7, the most down-regulated protein (0.039 fold change and Log_2_ Fold change −4.65302), directly represses the expression of specific G1/S genes involved in DNA replication, metabolism and DNA repair and, in certain tumor cell lines, particularly those derived from hematopoietic and lymphoid malignancies, E2F7 is expressed at very low levels [40]. It is noticeable that alongside the down expression of E2F7 and proteins such as KIF22 and KIF18A also involved in cell cycle and cell cycle process, we found four up-regulated proteins, HSP90AA1 and ORC2, which participate in the G1/S phase of the cell cycle, and HIST1H2BD and HIST1H2AH involved in nucleosome formation. This is particularly interesting due to the expression of HIST1H2AH, HSP90AA1, KIF18A and the same E2F7 is regulated by members of the E2F family.

E2F7 is a member of the E2F family of transcription factors that regulate the cell cycle, apoptosis, differentiation and senescence [41]. E2F is a key target for the retinoblastoma tumor suppressor pRb, and deregulation of the pathway is a frequent event in many different cancers and is thought to be a critical driver of uncontrolled proliferation in cancer cells where aberrant pRb activity occurs through a variety of oncogenic mechanisms [40,42]. E2F7 (as well as E2F8) is an atypical member of the E2F family because it is endowed with pRb-independent repressive activity, and it is known to be induced in late G1 by E2F1, together with an array of E2F target genes during the cell cycle [27]. E2F7 binds to promoters of microRNA and protein-coding genes bearing E2F consensus motifs, such as E2F1, CDC6, MCM2 or miR-25, during the S-phase, thereby repressing their expression. These findings have raised the possibility that E2F7 protein may be a key component of a negative feedback loop required to turn off transcription of E2F-driven G1/S target genes, thus allowing progression through the cell cycle. Accordingly, overexpression of E2F7 blocks S-phase entry, whereas acute loss of E2F7 accelerates cell-cycle progression [25,40,43,44]. In addition, in certain tumor cell lines, particularly those derived from hematopoietic and lymphoid malignancies and clinical diseases, such as glioblastoma, E2F7 is expressed at very low levels [40]. The degradation of atypical repressors E2F7 and E2F8 in late G2 resets the E2F transcriptional program just before cell division, presumably in preparation for the subsequent G1, in which the expression of E2F targets gradually increases and must reach a critical level in order to pass the restriction point and progress to the S phase [45]. Even more, the transcription of genes involved in DNA replication, metabolism and DNA repair are direct targets of E2F7, whereas mitotic or cytokinetic genes are not regulated by this factor [46,47].

E2F7 has also been involved in stress response because of depletion of atypical members of the E2F family; thus, E2F7 and E2F8 reduce the survival of tumor cells, primary mouse keratinocytes and embryonic fibroblasts after treatment with several DNA damaging compounds, indicating that sensitivity to cytotoxic/genotoxic stimuli is enhanced by loss of E2F7 or by the combined loss of E2F7/E2F8. Stress-induced apoptosis could be rescued when, under these circumstances, E2F1 is co-depleted [44]. This has accelerated tumorigenesis in a two-stage skin carcinogenesis model and confirmed a key role for E2F1 in E2F7/E2F8 dependent stress responses [46].

Moreover, E2F7 activity is associated with a suppression of DNA damage response (DDR) and DNA repair (DR) reactions. The expression of genes involved in DDR and DR pathways is cell cycle-regulated, showing a high gene expression in G1/S transition and decreasing thereafter. Deregulation of some of these genes throughout the cell cycle is E2F7-dependent [27,44]. Furthermore, it has also been shown that E2F7 binds to double-strand break sites and inhibits their repair, potentially by altering chromatin status at these sites [47]. Finally, Coutts et al. describe an unexpected and surprising role for E2F7 in regulating ribosomal gene transcription, providing evidence that links E2F7 activity with ribosomal biogenesis and a mechanism for integrating cell cycle progression with cell growth and protein synthesis [40].

In our study, we found proteins involved in cell cycle regulation whose genes are targets of the E2F family. The up-regulated proteins Hist1H2AH and HSP90AA1, which participate in the G1/S phase of the cell cycle, and the down-regulated proteins KIF18A and E2F7, that take part in the G2/M and the G1/S phase, respectively. Although further research will be needed to elucidate the molecular mechanisms underlying E2F7 participation in ALPS, our data show that this transcription factor can play a key role in the disease. Unfortunately, during this investigation, one patient’s medical condition worsened, and the attending doctors followed the protocol to alleviate her pain and practiced a splenectomy. Fortunately, the procedure worked in a good way for the girl. However, it disrupted the process of our work because it introduced a different situation in the analysis of the proteins, and we could not obtain more proteins to get the complete sequence of them or to perform other experiments.

## 4. Materials and Methods

### 4.1. Patients

Two patients with clinical evidence of ALPS were evaluated and underwent a review of medical history and records, physical examination, and routine laboratory studies. Blood samples were obtained, and peripheral blood mononuclear cells (PBMCs) were purified by Ficoll-Paque density centrifugation [48]. A sample from a healthy donor (a 13-year-old-girl) was used as a control, and, for molecular study, the sequences obtained also were compared to the reported in the Gene Bank. The authors declare that the procedures followed were in accordance with the regulations of the relevant clinical research ethics committee and with those of the Code of Ethics of the World Medical Association (Declaration of Helsinki). Parents provided a written informed consent form.

### 4.2. RNA Analysis

Total RNA from PBMCs was isolated using Trizol (Invitrogen, Carlsband, CA, USA) following standard protocol and quantified using an Epoch nanodrop spectrophotometer (Biotek, Winooski, VT, USA). Reverse transcription was performed on 0.4 µg RNA by one-step RT-PCR (Invitrogene, Carlsband, CA, USA), with specific primers for reactions that cover all the coding sequences of *Fas*, *FasL* and *Casp10* genes (Table 5). PCR conditions for cDNA amplification were denaturation at 94 °C for 40 s; annealing temperature varied according to the amplicon and extension at 72 °C 40–80 s for a total of 40 cycles. Aliquots of the amplified products were separated on a 1% agarose gel with ethidium bromide and visualized using a GelDoc (UVP) (Upland, CA, USA). The rest of the products were purified by using Centri-Sep columns (Princeton Separations, Adelphia, NJ, USA) following the manufacturer’s protocols and sequenced by Sanger’s dideoxy chain terminator method with dye-labeled dideoxy terminators (Applied Biosystems, Foster City, CA, USA); several primers to cover the total extension of the fragments were used (Sigma-Aldrich, St. Louis, MO, USA) (Table 5). Sequencing reaction products were purified with Centri-Sep columns and analyzed in an Applied Biosystems 310 Genetic Analyzer. Additionally, fragments of specific genes (Table 5) from some of the detected proteins were reverse transcribed, amplified, separated on agarose gels, and visualized on a GelDoc (UVP) (Upland, CA, USA). The fragment of E2F7 was also sequenced.

### 4.3. Label-Free Mass Spectrometry

#### 4.3.1. Sample Preparation

To isolate total proteins from PBMCs, we introduced a modification in the method described by Thadikkaran et al. [53]; that is, we disrupted the nuclei by the use of 40 cycles (30 s on/30 s off) in a Bioruptor (Diagenode, Denville, NJ, USA). Proteins were quantified by 2D Quant Kit (GE Healthcare, Chicago, IL, USA), and 50 µg for each condition were enzymatically digested as described [54]. Briefly, 10 µL of 50 mM ammonium bicarbonate were added to the samples, followed by the addition of 25 µL of 2% solution of RapiGest SF, (Waters, Milford, MA, USA); the tubes were heated for 15 min at 80 °C; then, 2.5 µL of 100 mM DTT (Sigma-Aldrich, St. Louis, MO, USA) were added, and a new incubation was carried out for 30 min at 60 °C. After the samples were cooled to room temperature, 2.5 µL of 300 mM Iodoacetamide (IAA) were added, leaving the samples 30 min in the dark at room temperature. After this, digestion with 500 ng of trypsin (Sigma-Aldrich, St. Louis, MO, USA) was performed at 37 °C overnight. Following the digestion, RapiGest was hydrolyzed with 10 µL of 5% trifluoroacetic acid (TFA) for 90 min at 37 °C. Samples were centrifugated at 14,000 rpm for 30 min at 6 °C. Finally, supernatants were transferred to Waters total recovery vials (Waters, Milford, MA, USA).

#### 4.3.2. Quantitative Mass Spectrometry and Data Analysis

After digestion, a spectrometric analysis was performed by using chromatographic and spectrometric conditions as it has been described [55]. Briefly, the resulting peptides were separated on HSS T3 C18 Column (Waters, Milford, MA, USA); 75 μm × 150 mm, 100 A° pore size, 1.8 μm particle size; using a UPLC ACQUITY M-Class using mobile phase A, 0.1% formic acid (FA) in H_2_O and mobile phase B, 0.1% FA in acetonitrile (ACN), under the following gradient: 0 min 7% B, 121.49 min 40% B, 123.15 to 126.46 min 85% B, 129 to 130 min 7% B, at a flow of 400 nL.min-1 and 45 °C. The spectra data were acquired in a Synapt G2-S*i* mass spectrometer (Waters, Milford, MA, USA) using a data-independent acquisition (DIA) approach through UDMS^E^ mode. Values for the ionization source were as follows: 2.75 kV in the sampler capillary, 30 V in the sampling cone, 30 V in the source offset, 70 °C for the source temperature, 0.5 Bar for the nanoflow gas and 150 L·h^−1^ for the purge gas flow. Low and high energy chromatograms were acquired in positive mode using *m*/*z* 50–2000 range with a scan time of 500 ms. No collision energy was applied to obtain the low energy chromatogram, while for the high energy chromatograms, the precursor ions were fragmented applying *quasi*-specific collision energies based on the drift time for each peptide detected by the mass spectrometer in UDMS^E^ mode. The control and sample were injected 3 times for statistical purposes.

The MS and MS/MS measurements contained in the generated *.raw files were normalized, aligned, identified according to Li et al. [56], compared and relatively quantified using Progenesis QI for Proteomics software *v*3.0.3 (Nonlinear Dynamics, Newcastle, UK) [57] against a *Homo sapiens* *.fasta database (downloaded from Uniprot, 71,758 protein sequences, last modified on 12 January 2017) concatenated with its reversed decoy database [58]. The identification of proteins includes cysteine carbamidomethylation as fixed modification, methionine oxidation and serine, threonine and tyrosine phosphorylation as variable modifications, trypsin as cut enzyme and one missed cleavage allowed; default peptide and fragment tolerance (maximum normal distribution of 10 ppm and 20 ppm, respectively) and false discovery rate ≤ 4%. Synapt G2-S*i* was calibrated as described [55] at 1.3 ppm across all MS/MS measurements.

Reliability and confidence for label-free experiments were verified at protein and peptide levels as described by Souza et al. [23]. To consider proteins differentially regulated, a ratio of ±1.2 (expressed as a base 2 logarithm) was calculated based on the average MS signal response of the three most intense tryptic peptides (Hi3) of each characterized protein in the patient (numerator) by an average of the Hi3 of each protein in the control (denominator). This means that these proteins had at least ±2.29 absolute fold change.

Data from the mass spectrometry experiment have been deposited to the ProteomeXchange Consortium via the PRIDE partner repository with accession identifier PXD013289 [59].

#### 4.3.3. Bioinformatics Analysis

Search Tool for the Retrieval of Interacting Genes (STRING) database v10.5 (available online: https://string-db.org/; *v*11.0 software, accessed on 12 December 2021) was used to construct the interactome of differentially regulated proteins following the next settings: *Homo sapiens* database; text mining, experiments, database, coexpression, neighborhood, gene fusion and co-occurrence as active interaction source and 0.4 as a confidence score. Pathway analysis was performed on the KEEG and REACTOME bioinformatics databases (available online: https://reactome.org/; *v*71 software accessed on 12 December 2021). Ontology of the biological process was done using the Protein Analysis Through Evolutionary Relationships (PANTHER) overrepresentation test (available online: http://pantherdb.org/; *v*17.0 software, accessed on 12 December 2021), FDR < 0.05%, (*p*) value < 0.05 and GO Ontology *Homo sapiens* database release 2018-09-08. The obtained data were further analyzed, and graphs were prepared using MS Excel.

#### 4.3.4. Western Blotting

In this study, 18 µg of protein isolated from PBMCs were electrophoresed on 12% SDS-PAGE. Separated proteins were electroblotted to Amersham Protran Premium 0.45 NC nitrocellulose membrane (GE Healthcare, Chicago, IL, USA) and subjected to immunorevelation. Incubation with specific primary antibodies was overnight at 4 °C. Blots were washed and incubated for 1 h at room temperature with horseradish peroxidase-conjugated secondary antibodies anti-mouse or anti-rabbit. Visualization of antibody-labeled protein bands was achieved using enhanced chemiluminescence as per manufacturer’s guidelines Western Lightning Plus-ECL (PerkinElmer Inc., Waltham, MA, USA) and quantified with ImageJ software (NIH, Bethesda, MD, USA). Graphics were performed with GraphPad Prism software (San Diego, CA, USA). Primary antibodies: anti E2F7 (1:1000, ab56022; Abcam, Waltham, MA, USA), anti-CRLF3 (1:500, sc-398388; Santa Cruz Biotechnology, Dallas, TX, USA). Anti-actine (mouse monoclonal, 1:500), was provided by Dr. Manuel Hernández, from the Cell Biology Department, CINVESTAV-IPN, México. Secondary antibodies: HRP-Goat anti-mouse IgG (1:7500, 115-035-003) and HRP-Goat anti-rabbit IgG (1:7500, 111-035-144) were purchased from Jackson ImmunoResearch, West Grove, PA, USA).

## 5. Conclusions

Due to the difficulty of making a precise diagnosis of the patients probably suffering from ALPS, we performed a molecular analysis of the genes proposed to be involved in the disease. In one of the patients, we could find a mutation in the *Fas* gene, which has been involved as a cause of ALPS. On the other hand, a proteomic analysis of the PBMCs of the patient without mutations in any of the analyzed genes shows deregulation in the level of expression of proteins which participate in, among other processes, cell cycle, regulation of cell cycle arrest and immune response. The high levels of proteins involved in the cell cycle and cell cycle regulation, as well as the reduced amount of E2F7, a negative regulator in these processes, can promote the increase of the activity of the genes involved in DNA replication and DNA repair and, therefore, the lymphocyte proliferation; this is because deregulation of E2F7 activity has been correlated with aberrant cell proliferation and in some instances cell death.

E2F7 participates in the attenuation of DNA repair through the repression of genes required for the timely regulation of DNA replication fork-associated damage repair. In the context of the patient, low expression of E2F7 protein would mean that regulation is altered and the transcriptional program that contributes to the regulation of DNA repair and genomic integrity is altered, and consequently, cell cycle progression is impaired.

## Figures and Tables

**Figure 1 ijms-23-05366-f001:**
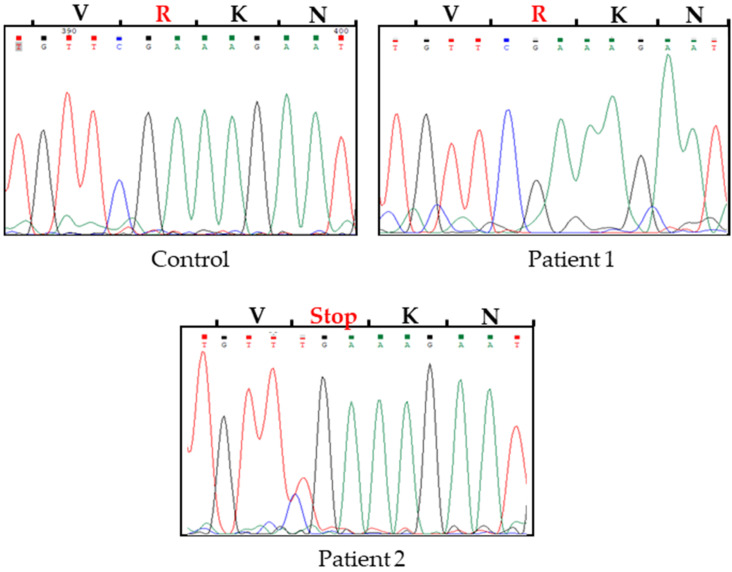
Molecular analysis of the sequence of *Fas*. mRNA was amplified by RT-PCR, and cDNA was sequenced directly. Electropherograms show a change of C to T identified in patient 2. As a control, we use the sample from a healthy 13-year-old girl.

**Figure 2 ijms-23-05366-f002:**
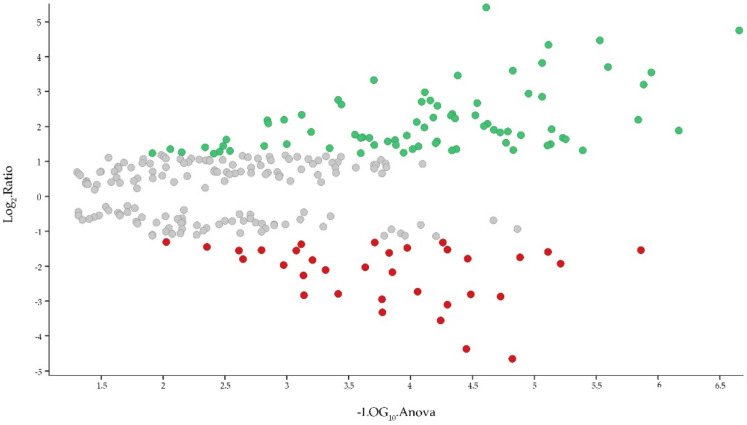
Volcano plot representing all filtered proteins. Gray circles unchanged proteins, green circles 77 up-regulated proteins, red circles 34 down-regulated proteins. *y* axis, ratio of the average of Hi3 intensities (ALPS-U/C) for each detected protein (values are represented as Log_2_). *x* axis, *p*-value of each detected protein in the triplicate (values represented as −Log_10_).

**Figure 3 ijms-23-05366-f003:**
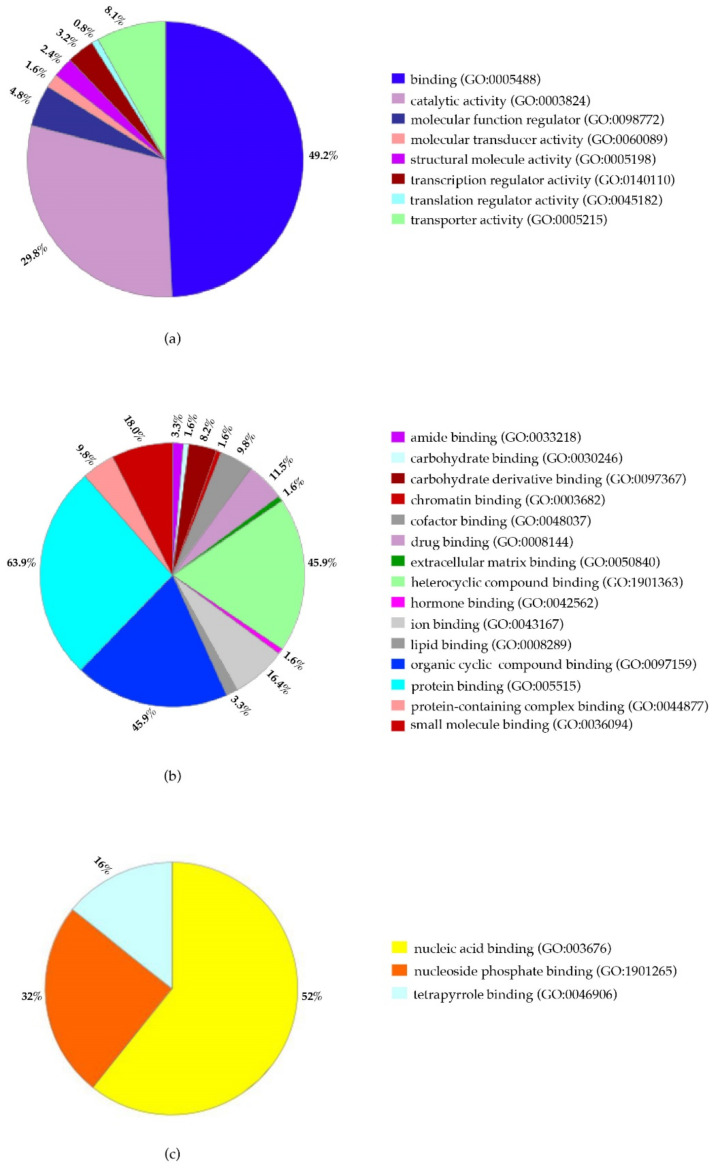
Classification of identified proteins. Gene ontology based on proteins involvement in (**a**), molecular function, (**b**), at binding level and (**c**), heterocyclic compound binding. Analysis performed by Panther Classification System.

**Figure 4 ijms-23-05366-f004:**
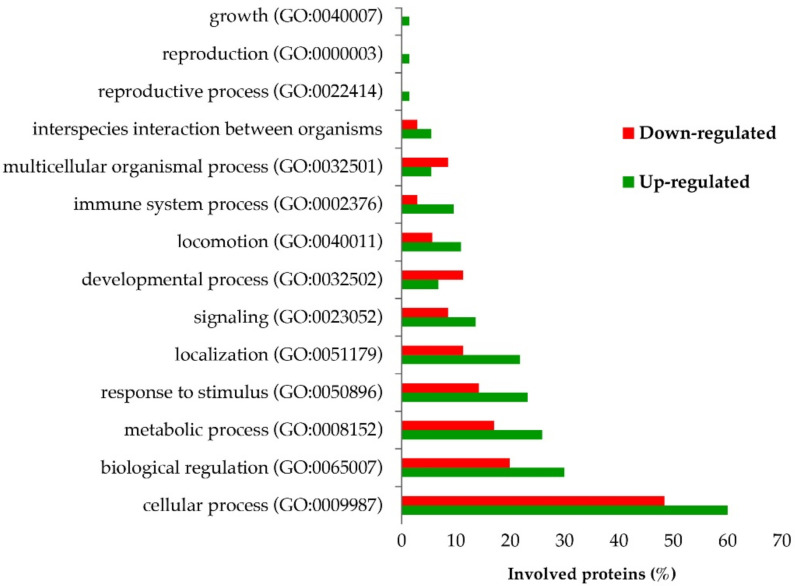
Gene ontology of differentially regulated proteins. Panther Classification System was used to analyze the identified proteins based on their involvement in biological processes.

**Figure 5 ijms-23-05366-f005:**
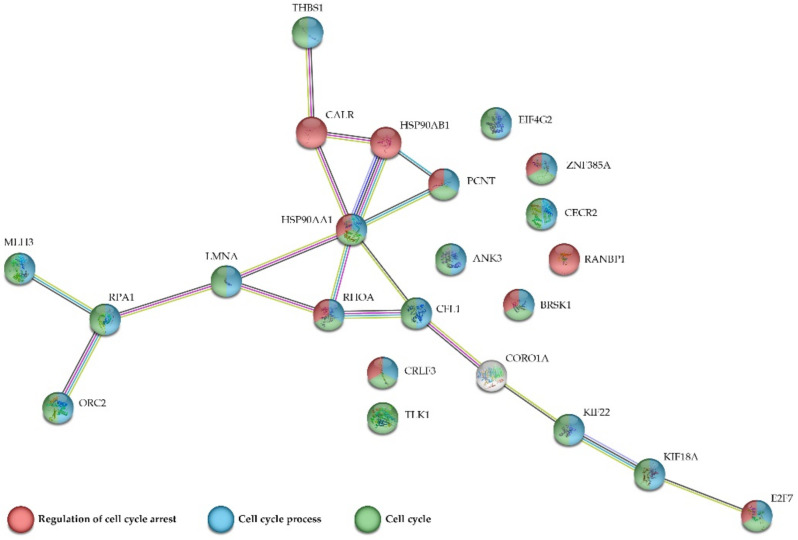
Network generated by the STRING database analysis. Interaction network of proteins involved in mentioned processes.

**Figure 6 ijms-23-05366-f006:**
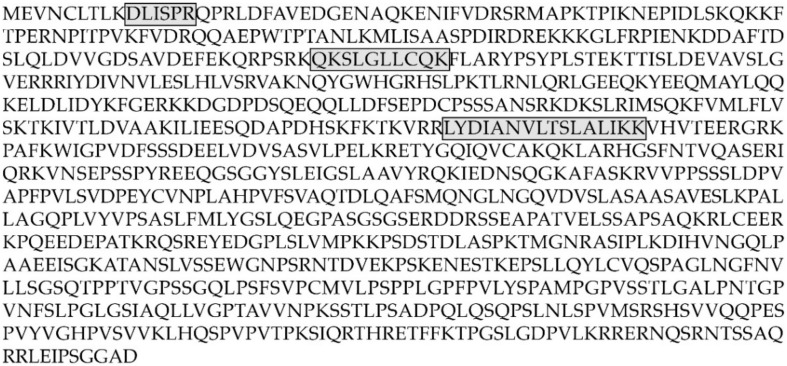
Amino acid sequence of E2F7 transcription factor (sp|Q96AV8|). Boxes show the peptides identified by label-free mass spectrometry.

**Figure 7 ijms-23-05366-f007:**
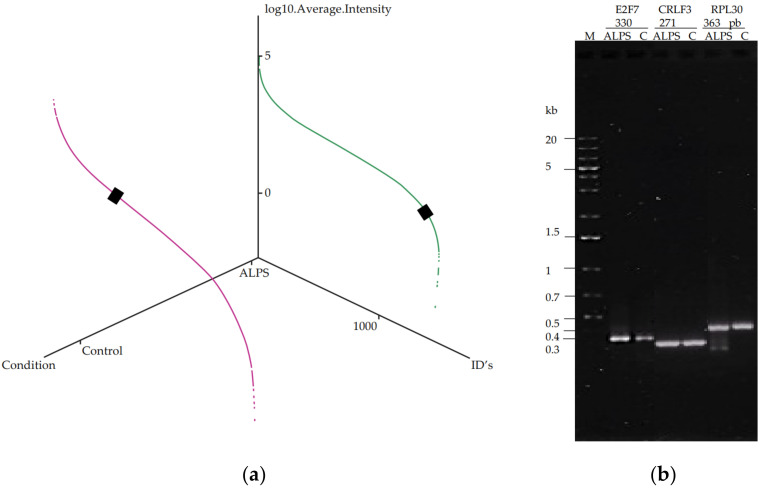
Molecular analysis of E2F7 in the ALPS-U patient. (**a**) Dynamic range of E2F7 expressed as base 10 logarithm. Magenta control, green ALPS-U samples. (**b**) Detection of E2F7 and CRLF3 by RT-PCR. RFLP30 was used as control.

**Figure 8 ijms-23-05366-f008:**
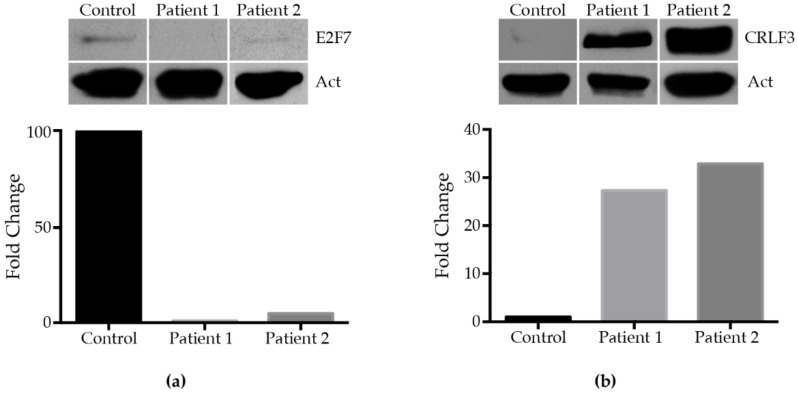
Immunodetection of (**a**) E2F7 and (**b**) CRLF3. In each case, immunodetection of actine was used as a control. Intensity of detected E2F7 and CRLF3 bands was quantified, and graphics performed with GraphPad Prism 6 software.

**Table 1 ijms-23-05366-t001:** Clinical and laboratory features of patients.

	Patient 1	Patient 2	Normal Range
Age (years)	13	6	
Age at diagnosis (years)	10	6	
Sex	Female	Male	
*Clinical data*			
Splenomegaly/Hepatomegaly	+	+	
Lymphadenopathy	+	+	
*Autoimmunity*			
Anemia	+	+	
Thrombopenia	+	+	
Neutropenia	+	+	
*Laboratory results*			
Lymphocyte count (cells/µL)	1450	3025	1800–2200
*T-cell* (%)			
CD3	88.5	77.9	60–85
CD4	81.2	43.2	29–59
CD8	5.28	28.9	19–48
DN	1.8	3.2	<1%
*B-cells* (%)			
CD19	-	12.7	8–20
CD27+	-	-	8–45
IgD-CD27+ (switching)	-	-	5–25
CD21Low	-	-	1.5–9
*Serum immunoglobulins*			
(mg/dl)			
IgG	2557	3596	400–1000
IgA	452	200	68–385
IgM	292	328	60–264
*Plasma biomarkers* (pg/mL)			
IL-10	-	-	
sFalL	-	-	
sCD25 (U/mL)	-	-	
Vitamin B12 (pg/mL)	-	1412	180–914
Treatment	Corticosteroids, spironolactone, ursodeoxycholic acid, azathioprine, Mycophenolate, Red Blood Cells transfusions	Corticosteroids, Mycophenolate	

**Table 2 ijms-23-05366-t002:** Amino acid sequence alignment of the canonical sequence of Fas (UniProtKB, P25445) and sequences from the patients. Stop codon in the sequence of patient 2 is showed.

Fas	MLGIWTLLPLVLTSVARLSSKSVNAQVTDINSKGLELRKTVTTVETQNLEGLHHDGQFCH	60
Patient_1	MLGIWTLLPLVLTSVARLSSKSVNAQVTDINSKGLELRKTVTTVETQNLEGLHHDGQFCH	60
Patient_2	MLGIWTLLPLVLTSVARLSSKSVNAQVTDINSKGLELRKTVTTVETQNLEGLHHDGQFCH	60
Fas	KPCPPGERKARDCTVNGDEPDCVPCQEGKEYTDKAHFSSKCRRCRLCDEGHGLEVEINCT	120
Patient_1	KPCPPGERKARDCTVNGDEPDCVPCQEGKEYTDKAHFSSKCRRCRLCDEGHGLEVEINCT	120
Patient_2	KPCPPGERKARDCTVNGDEPDCVPCQEGKEYTDKAHFSSKCRRCRLCDEGHGLEVEINCT	120
Fas	RTQNTKCRCKPNFFCNSTVCEHCDPCTKCEHGIIKECTLTSNTKCKEEGSRSNLGWLCLL	180
Patient_1	RTQNTKCRCKPNFFCNSTVCEHCDPCTKCEHGIIKECTLTSNTKCKEEGSRSNLGWLCLL	180
Patient_2	RTQNTKCRCKPNFFCNSTVCEHCDPCTKCEHGIIKECTLTSNTKCKEEGSRSNLGWLCLL	180
Fas	LLPIPLIVWVKRKEVQKTCRKHRKENQGSHESPTLNPETVAINLSDVDLSKYITTIAGVM	240
Patient_1	LLPIPLIVWVKRKEVQKTCRKHRKENQGSHESPTLNPETVAINLSDVDLSKYITTIAGVM	240
Patient_2	LLPIPLIVWVKRKEVQKTCRKHRKENQGSHESPTLNPETVAINLSDVDLSKYITTIAGVM	240
Fas	TLSQVKGFVRKNGVNEAKIDEIKNDNVQDTAEQKVQLLRNWHQLHGKKEAYDTLIKDLKK	300
Patient_1	TLSQVKGFVRKNGVNEAKIDEIKNDNVQDTAEQKVQLLRNWHQLHGKKEAYDTLIKDLKK	300
Patient_2	TLSQVKGFVStop	249
Fas	ANLCTLAEKIQTIILKDITSDSENSNFRNEIQSLV	335
Patient_1	ANLCTLAEKIQTIILKDITSDSENSNFRNEIQSLV	335
Patient_2		249

**Table 3 ijms-23-05366-t003:** Amino acid sequence alignment of Caspase 10 from the patients and the reported in UniProtKB–Q92851. *, amino acids which mutation has been implicated in ALPS, L285F [17], I406L [18], Y446C [14].

Caspase_10	MKSQGQHWYSSSDKNCKVSFREKLLIIDSNLGVQDVENLKFLCIGLVPNKKLEKSSSASD	60
Patient_1	MKSQGQHWYSSSDKNCKVSFREKLLIIDSNLGVQDVENLKFLCIGLVPNKKLEKSSSASD	60
Patient_2	MKSQGQHWYSSSDKNCKVSFREKLLIIDSNLGVQDVENLKFLCIGLVPNKKLEKSSSASD	60
Caspase_10	VFEHLLAEDLLSEEDPFFLAELLYIIRQKKLLQHLNCTKEEVERLLPTRQRVSLFRNLLY	120
Patient_1	VFEHLLAEDLLSEEDPFFLAELLYIIRQKKLLQHLNCTKEEVERLLPTRQRVSLFRNLLY	120
Patient_2	VFEHLLAEDLLSEEDPFFLAELLYIIRQKKLLQHLNCTKEEVERLLPTRQRVSLFRNLLY	120
Caspase_10	ELSEGIDSENLKDMIFLLKDSLPKTEMTSLSFLAFLEKQGKIDEDNLTCLEDLCKTVVPK	180
Patient_1	ELSEGIDSENLKDMIFLLKDSLPKTEMTSLSFLAFLEKQGKIDEDNLTCLEDLCKTVVPK	180
Patient_2	ELSEGIDSENLKDMIFLLKDSLPKTEMTSLSFLAFLEKQGKIDEDNLTCLEDLCKTVVPK	180
Caspase_10	LLRNIEKYKREKAIQIVTPPVDKEAESYQGEEELVSQTDVKTFLEALPQESWQNKHAGSN	240
Patient_1	LLRNIEKYKREKAIQIVTPPVDKEAESYQGEEELVSQTDVKTFLEALPQESWQNKHAGSN	240
Patient_2	LLRNIEKYKREKAIQIVTPPVDKEAESYQGEEELVSQTDVKTFLEALPQESWQNKHAGSN	240
Caspase_10	GNRATNGAPSLVSRGMQGASANTLNSETSTKRAAVYRMNRNHRGLCVIVNNHSFTSLKDR	300
Patient_1	GNRATNGAPSLVSRGMQGASANTLNSETSTKRAAVYRMNRNHRGLCVIVNNHSFTSLKDR	300
Patient_2	GNRATNGAPSLVSRGMQGASANTLNSETSTKRAAVYRMNRNHRGLCVIVNNHSFTSLKDR	300
	*	
Caspase_10	QGTHKDAEILSHVFQWLGFTVHIHNNVTKVEMEMVLQKQKCNPAHADGDCFVFCILTHGR	360
Patient_1	QGTHKDAEILSHVFQWLGFTVHIHNNVTKVEMEMVLQKQKCNPAHADGDCFVFCILTHGR	360
Patient_2	QGTHKDAEILSHVFQWLGFTVHIHNNVTKVEMEMVLQKQKCNPAHADGDCFVFCILTHGR	360
Caspase_10	FGAVYSSDEALIPIREIMSHFTALQCPRLAEKPKLFFIQACQGEEIQPSVSIEADALNPE	420
Patient_1	FGAVYSSDEALIPIREIMSHFTALQCPRLAEKPKLFFIQACQGEEIQPSVSIEADALNPE	420
Patient_2	FGAVYSSDEALIPIREIMSHFTALQCPRLAEKPKLFFIQACQGEEIQPSVSIEADALNPE	420
	*	
Caspase_10	QAPTSLQDSIPAEADFLLGLATVPGYV	447
Patient_1	QAPTSLQDSIPAEADFLLGLATVPGYV	447
Patient_2	QAPTSLQDSIPAEADFLLGLATVPGYV	447
	*	

**Table 4 ijms-23-05366-t004:** Amino acid sequence alignment of FasL from the patients and the reported in the UniProtKB–P48023. *, amino acids which mutation has been implicated in ALPS, P69A [19]; M86V, R156G [20]; A247E [21]; F275L [20]; G277S [13].

FasL	MQQPFNYPYPQIYWVDSSASSPWAPPGTVLPCPTSVPRRPGQRRPPPPPPPPPLPPPPPP	60
Patient_1	MQQPFNYPYPQIYWVDSSASSPWAPPGTVLPCPTSVPRRPGQRRPPPPPPPPPLPPPPPP	60
Patient_2	MQQPFNYPYPQIYWVDSSASSPWAPPGTVLPCPTSVPRRPGQRRPPPPPPPPPLPPPPPP	60
FasL	PPLPPLPLPPLKKRGNHSTGLCLLVMFFMVLVALVGLGLGMFQLFHLQKELAELRESTSQ	120
Patient_1	PPLPPLPLPPLKKRGNHSTGLCLLVMFFMVLVALVGLGLGMFQLFHLQKELAELRESTSQ	120
Patient_2	PPLPPLPLPPLKKRGNHSTGLCLLVMFFMVLVALVGLGLGMFQLFHLQKELAELRESTSQ	120
	* *	
FasL	MHTASSLEKQIGHPSPPPEKKELRKVAHLTGKSNSRSMPLEWEDTYGIVLLSGVKYKKGG	180
Patient_1	MHTASSLEKQIGHPSPPPEKKELRKVAHLTGKSNSRSMPLEWEDTYGIVLLSGVKYKKGG	180
Patient_2	MHTASSLEKQIGHPSPPPEKKELRKVAHLTGKSNSRSMPLEWEDTYGIVLLSGVKYKKGG	180
	*	
FasL	LVINETGLYFVYSKVYFRGQSCNNLPLSHKVYMRNSKYPQDLVMMEGKMMSYCTTGQMWA	240
Patient_1	LVINETGLYFVYSKVYFRGQSCNNLPLSHKVYMRNSKYPQDLVMMEGKMMSYCTTGQMWA	240
Patient_2	LVINETGLYFVYSKVYFRGQSCNNLPLSHKVYMRNSKYPQDLVMMEGKMMSYCTTGQMWA	240
FasL	RSSYLGAVFNLTSADHLYVNVSELSLVNFEESQTFFGLYKL	281
Patient_1	RSSYLGAVFNLTSADHLYVNVSELSLVNFEESQTFFGLYKL	281
Patient_2	RSSYLGAVFNLTSADHLYVNVSELSLVNFEESQTFFGLYKL	281
	* * *	

**Table 5 ijms-23-05366-t005:** Oligonucleotide primers used in this study.

Primer	Sequence	Reference
*Fas*		
FasA	5′-AAGCTCTTTCACTTCGGAGG3′	[37]
FasRL	5′-CAATGTGTCATACGCTTCT-3′	[37]
FasE	5′-AGGACATGGCTTAGAAGTG-3′	[37]
FasRN	5′-ACAGCCAGCTATTAAGAAT-3′	[37]
*FasL*		
FasLs	5′-TAAAACCGTTTGCTGGGGCT-3′	This work
FasLas	5′-TCGGAGTTCTGCCAGCTCCTT-3′	This work
FasLasb	5′-ATTGAACACTGCCCCCAGGT-3′	This work
FasL423	5′-AAGGAGCTGGCAGAACTCCGA-3′	This work
FasL757	5′AGGATCTGGTGATGATGGAG-3′	This work
FasL3′ Del Rey	5′-GAGAAGCACTTTGGGATTCTTTCC-3	[21]
*Casp10*		
Ash	5′-CCATGAAATCTCAAGGTCAACATTGG-3′	[49]
WangV7R	5′-GCATAGTCTTCAGGTGGGCGTTTG-3′	[16]
319F	5′GTGGAGCGACTGCTGCCCACCCGA3′	This work
592F	ATCCAGATAGTGACACCTCCTGTA3′	This work
e9bs	5‘-CGAAAGTGGAAATGGAGATGGT-3′	[50]
e9cas	5′-CCACATGCCGAAAGGATACA-3′	[50]
*E2F7*		
E2F7forClements	5′-GGAAAGGCAACAGCAAACTCT-3′	[51]
E2F7revClements	5′-TGGGAGAGCACCAAGAGTAGAAGA-3′	[51]
*CRLF3*		
For_CRLF3_Yang	5′-AACGTTGATTACCAGTTCAG-3′	[52]
Rev_CRLF3_Yang	5′-CTGAGGACAGCTACGTTAGA-3′	[52]
*MLH3*		
MLH3DMFor721	5′-GAGAAGGTTAGGCAGAGAATA-3′	This work
RC1141	5′-ACAGAATTGGCACTGCACATT-3′	This work
*RPL30* F-RPL30 R-RPL30		
	5′-CTCCCAAAGGCTATTCAGTAATGG-3′	Tapia-Ramírez J., personal communication
	5′-GCTAAAAGGTGCTCGCTTCAGC-3′	

## Data Availability

Data from the mass spectrometry experiment have been deposited to the ProteomeXchange Consortium via the PRIDE partner repository with accession identifier PXD013289.

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
