# Peer review of "Differential Expression of Proteins in an Atypical Presentation of Autoimmune Lymphoproliferative Syndrome"

_ijms, 2022, doi:10.3390/ijms23105366_

Round 1
Reviewer 1 Report
The resubmitted manuscript "Differential expression of proteins in an atypical presentation of Autoimmune Lymphoproliferative Syndrome" has been significantly improved. The Authors modified the text according to all my suggestions and comments.
Author Response
Thank you for your valuable comments to improve our manuscript
Reviewer 2 Report
The manuscript is interesting, the application of modern techniques, including proteomics, in studies on rare disorders is of great importance for finding the molecular mechanism of the disease and for potentiaÈ™ therapeutic approaches. I recommend minor revision.

Author Response
Thank you for your valuable comments to improve our manuscript

This manuscript is a resubmission of an earlier submission. The following is a list of the peer review reports and author responses from that submission.
Round 1
Reviewer 1 Report
This study by Delgadillo et al., investigates the differential protein expression in peripheral blood mononuclear cells between one patient with undetermined Autoimmune Lymphoproliferative Syndrome (U-ALPS) and a healthy control. Label-free quantitative proteomics was used to compare both samples, measured in triplicates. Of 1399 proteins, as it is said in the text (although in the provided excel file appear 1522), 266 showed significant changes. Enrichment analysis suggest proteins involved in cell cycle, cell cycle regulation, immune response, apoptosis, and regulation of cell death. Based on the proteins identified it is suggested an alteration in cell cycle and apoptotic cell death.
Comparing proteomes from patients and healthy donors is a worthy approach. However, in this case the study lack of reliability due to sample size (only one of each) and the available amount of protein (only a replicate in the western blots). It would be a good idea to include in the study at least a sample from the patient with genetic ALPS to identify proteins altered in a general way common to this pathology no matter the origin. All the conclusions are drawn from the matches between proteomics candidates and gene signatures linked to cell cycle, apoptosis, and immune response. Very little is done to address the functional importance of candidates in U-ALPS sample and the interrelation and pathways in which they are involved. For instance, proteins related to cell cycle are both up and downregulated, how are they related? So, manuscript is formulated in such a way that results are merely descriptive. In addition, the methods used to conduct proteomics are incompletely described, and the results are displayed in a format that is difficult to understand (for example, adding an excel only with the proteins that change significantly would be appreciated) and some of them, as tables 5 and 6, must be supplementary material. I found difficult to follow authors logic, especially in the results section (Figure 7b, expression of E2F7 seems to be decreased, but authors said that no differences were found), and there are also errors in some figures (Figure 8b, western of CRLF3 but in the image appears E2F7). Finally, the conclusions are contradictory. Authors claim the E2F7 (a negative regulator of cell cycle) is downregulated in U-ALPS which activates cell cycle and may trigger apoptosis. However, the basis of ALPS is a defect in the lymphocyte apoptotic pathway. I´m also concern about the results being due to the treatment and not to pathology in the patient. In summary, the manuscript should be improved by performing additional experiments (like determine cell proliferation and cell cycle phases by flow cytometry, since the main alterations founded are related to cell cycle) and revising the main text.
Reviewer 2 Report
In the manuscript “Differential expression of proteins in an atypical presentation of Autoimmune Lymphoproliferative Syndrome” the Authors performed a molecular and proteomic analysis of peripheral blood mononuclear cells from two patients with a clinical diagnosis of autoimmune lymphoproliferative syndrome (ALPS). ALPS is a rare disease commonly caused by mutations in genes affecting the extrinsic apoptotic pathway. However, up to 30% of patients clinically diagnosed do not present any defined mutations. Prompt and correct recognition of ALPS is important to avoid a false diagnosis and guarantee an adequate treatment of the disease.
Although the subject is interesting and methods chosen for the study were appropriate, some concerns need to be addressed before the manuscript is ready for publication.
Comments:
- Abstract should be modified (lines 20-28). I suggest to avoid the description of patients as “patient 1” and “patient 2” without providing more detailed data. I highly recommend to address this issue. Please compare with the Sections 2.1. Clinical features of the patients and 2.3. Proteomic analysis of RBMCs.
- Introduction: I suggest to include references providing more current knowledge on ALPS.
- References should be numbered in order of appearance in the text. References in the Section 2. Results start from [23] instead of [13]. References numbered from [13] to [22] are included in the Section 5. Materials and Methods at the end of the manuscript.
- I suggest to use “PBMCs” instead of “RBMCs” as the acronym for peripheral blood mononuclear cells throughout the manuscript.
- Page 1, line 36: It should be “the tumor necrosis factor receptor I (TNFR I)” instead of “the tumor necrosis factor receptor I (TNFR)”.
- Page 4, Figure 1: Control should be described. Was it the same sample for the patient 1 (a 13-year-old girl) and patient 2 (a 6-year-old boy)?
- Page 4, line 105: I suggest “Electropherograms show a change” instead of “Electropherograms shown a change”. COOH terminal”
- Page 5, line 109: I suggest “in the COOH terminus” instead of “in the COOH terminal”.
- Table 3: Symbols * for L285 and Y446 should be placed more precisely.
- Table 4 title: I suggest to list amino acids which mutations has been implicated in ALPS in the order in which they appear in the FasL sequence (starting from P69A).
- Page 6, line 129: Acronyms should be defined the first time they appear in the text.
- Page 7,Table 5, second row: It should be “Log2 Fold change” instead of “Log2 Fold change”.
- Page 10, Table 5, Accession P49591: It should be “synthetase” instead of “sinthetase”.
- Page 11, Table 5, Accession P58876: It should be “Component” instead of “component” and “plays a central role” instead of “play a central role”.
- Page 13, Table 5, Accession Q93009-3: It should be “proteasomal degradation” instead of “proteosomal degradation”. The proteasome is the most complex protease known in eukaryotes. Its principal function is to degrade ubiquitin–protein conjugates.
- Page 14, Table 5: “DOWN” should be written using the same font type as for “UP”.
- Page 14, Table 5, Accession P02787: I suggest “transports iron” instead of “transport iron”.
- Page 14, Table 5, Accession Q9Y446-2: I suggest “Participates” instead of “Participate”.
- Page 15, Table 5, Accession P78344: I suggest “Plays” instead of “Play”.
- Page 15, Table 5, Accession Q12955: I suggest “Links” and “plays” instead of “Link” and “play”, respectively.
- Page 16, Table 5: No protein accession number is given for transcription factor E2F7.
- Page 16, Table 6, Accession P37235: It should be “signaling” instead of “signalling”. The Authors use “signaling” throughout the manuscript.
- Page 18, Table 6: “ALPS” should be written using the same font type as for “CONTROL”.
- Page 19, lines 160-161: I suggest “from the analyzed patient (ALPS-U)” instead of “from the analyzed patient”.
- Page 21, Figure 3a legend: It should be “catalytic activity (GO:0003824)” instead of “catalyticactivity(GO:0003824)”.
- Page 21, Figure 3a legend: It should be “translation” instead of “Translation”.
- Page 22, line 204: I suggest “in biological processes” instead of “in biological process”.
- Page 23, line 252: I suggest “in these mentioned processes” instead of “in these mentioned process”.
- Page 24, Figure 5 caption, line 269: I suggest “in mentioned processes” instead of “in mentioned process”.
- Page 24, line 294: It should be “which” instead of “wich”.
- Page 24, line 295: It should be “peptides” instead of “preptides”.
- Page 24, line 297: I suggest “the patient 1 (ALPS-U) compared to the control” instead of “the patient compared to the control”.
- Page 25, Figure 7 caption, lines 309-310: It should be “(a)” and “(b)” instead of “a,” and “b,”, respectively. See Figure 3 and Figure 8 captions for comparison.
- Page 27, line 362: I suggest “these two processes” instead of “these two-process”.
- Page 27, line 370: It should be “2.48 fold” instead of “1.31 fold”. For THBS1 the Authors used Log2 Fold change (1.312232) instead of Fold change (2.483254).
- Page 27, line 372: It should be “Log2 Fold change” instead of “Log2 Fold change”.
- Page 28, line 416: It should be “inhibits” instead of “inhibit”.
- Page 28, line 423: I suggest “this transcription factor can play a key role” instead of “this transcription factor can be a key role”.
- Page 28, line 427: I suggest “disrupted” instead of “disrupt”.
- Page 28, line 449: Control should be described more precisely. Compare with the comment No. 5.
- Page 29, Table 7: An appropriate reference number should be given instead of “(Wang et al., 2007)”. See lines 641-642.
- Page 30, line 482: I suggest “10 µl of 50 mM” instead of “10 µl 50 mM”.
- Page 30, line 483: I suggest “25 µl of 2%” instead of “25 µl 2%”.
- Page 30, line 484: I suggest “2.5 µl of 100 mM” instead of “2.5 µl 100 mM”.
- Page 34, line 662: It should be “1998” instead of “1998”.
- Page 34, line 667: it should be “2009” instead of “2009”.
- Extra spaces between words should be removed (e.g. lines 407, 481, 497, 501, 505).
- Text editing is recommended.

Author Response
Please see the atrtachment

Round 2
Reviewer 1 Report
I would like to thank the authors for their answer to my suggestions and comments. However, although more bibliographic information has been added, no significant improvement about the results has been made, since it has not brought any new results. I perfectly understand the complexity of work with patients, even more when we speak about a rare disease, but the conclusions on this work might not be accurate based on information provided. I suggested, for instance, to include in the study a sample from the patient with genetic ALPS to have more detailed information.
Further, when I said that “all the conclusions are drawn from the matches between proteomics candidates and gene signatures linked to cell cycle, apoptosis, and immune response”, I didn´t mean why you chose specific biological processes, but the fact that validation of these results is needed (cell cycle and apoptosis analysis and determination of proteins related to immune response).
When I refer that little is done to address the functional importance of candidates, I´m not expecting more information about them independently but more an interrelation between them.
I appreciate the improvement in the description about methodology used for proteomic analysis, the clarification of the number of identified and quantified proteins and the simplification of the manuscript moving some tables to supplementary material.
I know that gene and protein expression doesn´t correlate but if so, this must be clearly reflected in the text.
Regarding to my last comment, I agreed with the authors that based in proteomics a deregulation on cell cycle seems to happen in the patient but the consequences of this (increased or decreased lymphocyte proliferation, or cell cycle arrest) are not clear. Besides, I stand by my concern regarding the effects of the treatment. As stated by the authors, one of the drugs used in the treatment (mycophenolate) affects apoptosis through caspase 3 activation. However, caspase 3 seems decreased in ALPS condition, but again, without validation by western blot for example, it could not be assumed the activation state of caspase 3. As happen with gene and protein expression, less protein doesn´t mean less activity.